# Accelerated Projected Gradient Algorithms for Sparsity Constrained Optimization Problems

**Jan Harold Alcantara**
Academia Sinica
Taipei, Taiwan
jan.harold.alcantara@gmail.com

**Ching-pei Lee**
Academia Sinica
Taipei, Taiwan
leechingpei@gmail.com

## Abstract

We consider the projected gradient algorithm for the nonconvex best subset selection problem that minimizes a given empirical loss function under an $\ell_0$-norm constraint. Through decomposing the feasible set of the given sparsity constraint as a finite union of linear subspaces, we present two acceleration schemes with global convergence guarantees, one by same-space extrapolation and the other by subspace identification. The former fully utilizes the problem structure to greatly accelerate the optimization speed with only negligible additional cost. The latter leads to a two-stage meta-algorithm that first uses classical projected gradient iterations to identify the correct subspace containing an optimal solution, and then switches to a highly-efficient smooth optimization method in the identified subspace to attain superlinear convergence. Experiments demonstrate that the proposed accelerated algorithms are magnitudes faster than their non-accelerated counterparts as well as the state of the art.

## 1 Introduction

We consider the sparsity-constrained optimization problem in $\Re^n$:

$$\min_{w \in A_s} f(w), \tag{1}$$

where $f$ is convex with $L$-Lipschitz continuous gradient, $s \in \mathbb{N}$, and $A_s$ is the sparsity set given by

$$A_s \coloneqq \{w \in \Re^n : \|w\|_0 \leq s\}, \tag{2}$$

where $\|w\|_0$ denotes the $\ell_0$-norm that indicates the number of nonzero components in $w$. We further assume that $f$ is lower-bounded on $A_s$.

A classical problem that fits in the framework of (1) is the best subset selection problem in linear regression [6, 20]. Given a response vector $y \in \Re^m$ and a design matrix of explanatory variables $X \in \Re^{m \times n}$, traditional linear regression minimizes a least squares (LS) loss function

$$f(w) = \|y - Xw\|^2/2. \tag{3}$$

However, due to either high dimensionality in terms of the number of features $n$ or having significantly fewer instances $m$ than features $n$ (i.e., $m \ll n$), we often seek a linear model that selects only a subset of the explanatory variables that will best predict the outcome $y$. Towards this goal, we can solve (1) with $f$ given by (3) to fit the training data while simultaneously selecting the best-$s$ features. Indeed, such a sparse linear regression problem is fundamental in many scientific applications, such as high-dimensional statistical learning and signal processing [22]. The loss in (3) can be generalized to the following linear empirical risk to cover various tasks in machine learning beyond regression

$$f(w) = g(Xw), \quad g(z) = \sum_{i=1}^{m} g_i(z_i), \tag{4}$$

where $g$ is convex. Such a problem structure makes evaluations of the objective and its derivatives highly efficient, and such efficient computation is a key motivation for our algorithms for (1).

36th Conference on Neural Information Processing Systems (NeurIPS 2022).

**Related Works.** The discontinuous cardinality constraint in (1) makes the problem difficult to solve. To make the optimization problem easier, a popular approach is to slightly sacrifice the quality of the solution (either not strictly satisfying the sparsity level constraint or the prediction performance is deteriorated) to use continuous surrogate functions for the $\ell_0$-norm, which lead to a continuous nonlinear programming problem, where abundant algorithms are at our disposal. For instance, using a convex penalty surrogate such as the $\ell_1$-norm in the case of LASSO [36], the problem (1) can be relaxed into a convex (unconstrained) one that can be efficiently solved by many algorithms. Other algorithms based on continuous nonconvex relaxations such as the use of smoothly clipped absolute deviation [15] and the minimax concave penalty [41] regularizers are also popular in scenarios with a higher level of noise and outliers in the data. However, for applications in which enforcing the constraints or getting the best prediction performance is of utmost importance, solving the original problem (1) is inevitable. (For a detailed review, we refer the interested reader to [11, Section 1].) Unfortunately, methods for (1) are not as well-studied as those for the surrogate problems. Moreover, existing methods are indeed still preliminary and too slow to be useful in large-scale problems often faced in modern machine learning tasks.

In view of the present unsatisfactory status for scenarios that simultaneously involve high-volume data and need to get the best prediction performance, this work proposes efficient algorithms to directly solve (1) with large-scale data. To our knowledge, all the most popular algorithms that directly tackles (1) without the use of surrogates involve using the well-known projected gradient (PG) algorithm, at least as a major component [10–13, 3].[1] [10] proved linear convergence of the objective value with the LS loss function (3) for the iterates generated by PG under a scalable restricted isometry property, which also served as their tool to accelerate PG. However, given any problem instance, it is hard, if not computationally impossible, to verify whether the said property holds. On the other hand, [11] established global subsequential convergence to a stationary point for the iterates of PG on (1) without the need for such isometry conditions, and their results are valid for general loss functions $f$ beyond (3). While some theoretical guarantees are known, the practicality of PG for solving (1) remains a big problem in real-world applications as its empirical convergence speed tends to be slow. The PG approach is called iterative hard thresholding (IHT) in studies of compressed sensing [13] that mainly focuses on the LS case. To accelerate IHT, several approaches that alternates between a PG step and a subspace optimization step are also proposed [12, 3], but such methods mainly focus on the LS case and statistical properties, while their convergence speed is less studied from an optimization perspective. Recently, "acceleration" approaches for PG on general nonconvex regularized problems have been studied in [27, 37]. While their proposed algorithms are also applicable to (1), the obtained convergence speed for nonconvex problems is not faster than that of PG.

This work is inspired by our earlier work [1], which considered a much broader class of problems without requiring convexity nor differentiability assumptions for $f$, and hence obtained only much weaker convergence results, with barely any convergence rates, for such general problems.

**Contributions.** In this work, we revisit the PG algorithm for solving the general problem (1) and propose two acceleration schemes by leveraging the combinatorial nature of $\ell_0$-norm. In particular, we decompose the feasible set $A_s$ as the finite union of $s$-dimensional linear subspaces, each representing a subset of the coordinates $\{1, \ldots, n\}$, as detailed in (7) of Section 2. Such subspaces are utilized in devising techniques to efficiently accelerate PG. Our first acceleration scheme is based on a same-space extrapolation technique such that we conduct extrapolation only when two consecutive iterates $w_{k-1}$ and $w_k$ lie in the same subspace, and the step size for this extrapolation is determined by a spectral initialization combined with backtracking to ensure sufficient function decrease. This is motivated by the observation that for (4), objective and derivatives at the extrapolated point can be inferred efficiently through a linear combination of $Xw_{k-1}$ and $Xw_k$. The second acceleration technique starts with plain PG, and when consecutive iterates stay in the same subspace, it begins to alternate between a full PG step and a truncated Newton step in the subspace to obtain superlinear convergence with extremely low computational cost. Our main contributions are as follows:

1. We prove that PG for (1) is globally convergent to a local optimum with a local linear rate, improving upon the sublinear results of Bertsimas et al. [11]. We emphasize that our framework, like [11], is applicable to general loss functions $f$ satisfying the convexity and smoothness

---

[1] [17] proposed an algorithm for a similar optimization problem that minimizes $f(w) + C\|w\|_0$ for some $C > 0$. But whether it is equivalent to (1) is unclear because both problems are nonconvex, and for any prespecified sparsity level $s$, it is hard to find $C$ that leads to a solution $w^*$ with $\|w^*\|_0 = s$.

requirements, and therefore covers not only the classical sparse regression problem but also many other ones encompassed by the empirical risk minimization (ERM) framework.

2. By decomposing $A_s$ as the union of linear subspaces, we further show that PG is provably capable of identifying a subspace containing a local optimum of (1). By exploiting this property, we propose two acceleration strategies with practical implementation and convergence guarantees for the general problem class (1). Our acceleration provides both computational and theoretical advantages for convergence, and can in particular obtain superlinear convergence.

3. In comparison with existing acceleration methods for nonconvex problems [27, 37], this work provides new acceleration schemes with faster theoretical speeds (see Theorems 3.2 and 3.3), and beyond being applied to the classical PG algorithm, those schemes can also easily be combined with existing accelerated PG approaches to further make them converge even faster.

4. Numerical experiments exemplify the significant improvement in both iterations and running time brought by our acceleration methods, in particular over the projected gradient algorithm by [11] as well as the accelerated proximal gradient method for nonconvex problems proposed by [27].

This work is organized as follows. We review the projected gradient algorithm and prove its local linear convergence and subspace identification for arbitrary smooth loss functions in Section 2. In Section 3, we propose the acceleration schemes devised through decomposing the constraint set in (1) into subspaces of $\Re^n$. Experiments in Section 4 then illustrate the effectiveness of the proposed acceleration techniques, and Section 5 concludes this work. All proofs, details of the experiment settings, and additional experiments are in the appendices.

## 2 Projected Gradient Algorithm

The *projected gradient algorithm* for solving (1) is given by the iterations

$$w^{k+1} \in T_{\mathrm{PG}}^\lambda(w^k) := P_{A_s}(w^k - \lambda \nabla f(w^k)), \tag{5}$$

where $P_{A_s}(w)$ denotes the projection of $w$ onto $A_s$, which is set-valued because of the nonconvexity of $A_s$. When $f$ is given by (3), global linear convergence of this algorithm under a restricted isometry condition is established in [10]. For a general convex $f$ with $L$-Lipschitz continuous gradients, that is,

$$\|\nabla f(w) - \nabla f(w')\| \leq L\|w - w'\| \quad \forall w, w' \in \Re^n, \tag{6}$$

the global subsequential convergence of (5) is proved in [11], but neither global nor local rates of convergence is provided. In this section, we present an alternative proof of global convergence and more importantly establish its local linear convergence.

A useful observation that we will utilize in the proofs of our coming convergence results is that the nonconvex set $A_s$ given by (2) can be decomposed as a finite union of subspaces in $\Re^n$:

$$A_s = \bigcup_{J \in \mathcal{J}_s} A_J, \quad A_J := \mathrm{span}\{e_j : j \in J\}, \quad \mathcal{J}_s := \{J \subseteq \{1, 2, \ldots, n\} : |J| = s\}, \tag{7}$$

where $e_j$ is the $j$th standard unit vector in $\Re^n$. Throughout this paper, we assume that $\lambda \in (0, L^{-1})$.

**Theorem 2.1.** *Let $\{w^k\}$ be a sequence generated by* (5). *Then:*

*(a) (Subsequential convergence) Either $\{f(w^k)\}$ is strictly decreasing, or there exists $N > 0$ such that $w^k = w^N$ for all $k \geq N$. In addition, any accumulation point $w^*$ of $\{w^k\}$ satisfies $w^* \in P_{A_s}(w^* - \lambda \nabla f(w^*))$, and is hence a stationary point of* (1).

*(b) (Subspace identification and full convergence) There exists $N \in \mathbb{N}$ such that*

$$\{w^k\}_{k=N}^\infty \subseteq \bigcup_{J \in \mathcal{I}_{w^*}} A_J, \qquad \mathcal{I}_{w^*} := \{J \in \mathcal{J}_s : w^* \in A_J\}. \tag{8}$$

*whenever $w^k \to w^*$. In particular, if $T_{\mathrm{PG}}^\lambda(w^*)$ is a singleton for an accumulation point $w^*$ of $\{w^k\}$, then $w^*$ is a local minimum for* (1), *$w^k \to w^*$, and* (8) *holds.*

*(c) (Q-linear convergence) If $T_{\mathrm{PG}}^\lambda(w^*)$ is a singleton for an accumulation point $w^*$ and $w \mapsto w - \lambda \nabla f(w)$ is a contraction over $A_J$ for all $J \in \mathcal{I}_{w^*}$, then $\{w^k\}$ converges to $w^*$ at a Q-linear rate. In other words, there is $N_2 \in \mathbb{N}$ and $\gamma \in [0, 1)$ such that*

$$\|w^{k+1} - w^*\| \leq \gamma \|w^k - w^*\|, \quad \forall k \geq N_2. \tag{9}$$

It is well-known that an optimal solution of (1) is also a stationary point of it [8, Theorem 2.2], and therefore (a) proves the global subsequential convergence of PG to candidate solutions of (1). Consider $z^* := w^* - \lambda \nabla f(w^*)$, and let $\tau$ be a permutation of $\{1, \ldots, n\}$ such that $z^*_{\tau(1)} \geq z^*_{\tau(2)} \geq \cdots \geq z^*_{\tau(n)}$. The requirement of $T^\lambda_{\mathrm{PG}}(w^*)$ being a singleton in Theorem 2.1 (b) then simply means the mild condition of $z^*_{\tau(s)} > z^*_{\tau(s+1)}$, which is almost always true in practice. The requirement for (c) can be fulfilled when $f$ confined to $A_J$ is strongly convex, even if $f$ itself is not. This often holds true in practice when $f$ is of the form (4) and we restrict $s$ in (1) to be smaller than the number of data instances $m$, and is thus also mild. The existence of a stationary point can be guaranteed when $\{w^k\}$ is a bounded sequence, often guaranteed when $f$ is coercive on $A_J$ for each $J \in \mathcal{J}_s$.

In comparison to existing results in [11, 2, 14], parts (b) and (c) of Theorem 2.1 are new. In particular, part (b) provides a full convergence result that usually requires stronger regularity assumptions like the Kurdyka-Łojasiewicz (KL) condition [2, 14] (see also (21)) that requries the objective value to decrease proportionally with the minimum-norm subgradient in a neighborhood of the accumulation point, but we just need the very mild singleton condition right at the accumulation point only. Part (c) gives a local linear convergence for the PG iterates even if the problem is nonconvex, while the rates in [14] requires a KL condition and the rate is measured in the objective value.

The following result further provides rates of convergence of the objective values even without the conventional KL assumption. The first rate below follows from [24].

**Theorem 2.2.** *Let $\{w^k\}$ be a sequence generated by (5). If $w^k \to w^*$, such as when $T^\lambda_{\mathrm{PG}}(w^*)$ is a singleton at an accumulation point $w^*$ of (5), then*

$$f(w^k) - f(w^*) = o(k^{-1}). \tag{10}$$

*Moreover, under the hypothesis of Theorem 2.1 (c), the objective converges to $f(w^*)$ R-linearly, i.e.,*

$$f(w^k) - f(w^*) = O(\exp(-k)). \tag{11}$$

By using Theorem 2.1, we can also easily get rates faster than (10) under a version of the KL condition that is easier to understand and verify than those assumed in existing works. In particular, existing analyses require the KL condition to hold in a neighborhood in $\Re^n$ of an accumulation point, but we just need it to hold around $w^*$ within $A_J$ for the restriction $f|_{A_J}$ for each $J \in \mathcal{I}_{w^*}$. These results are postponed to Theorem 3.2 in the next section as the PG method is a special case of our acceleration framework.

## 3 Accelerated methods

The main focus of this work is the proposal in this section of new techniques with solid convergence guarantees to accelerate the PG algorithm presented in the preceding section. Our techniques fully exploit the subspace identification property described by the inclusion (8), as well as the problem structure of (4) to devise efficient algorithms.

We emphasize that the two acceleration strategies described below can be combined together, and they are also widely applicable such that they can be employed to other existing algorithms for (1) as long as such algorithms have a property similar to (8).

### 3.1 Acceleration by extrapolation

Traditional extrapolation techniques are found in the realm of convex optimization to accelerate algorithms [9, 31] with guaranteed convergence improvements, but were often only adopted as heuristics in the nonconvex setting, until some recent works showed that theoretical convergence can also be achieved [27, 37]. However, unlike the convex case, these extrapolation strategies for nonconvex problems do not lead to faster convergence speed nor an intuitive reason for doing so. An extrapolation step proceeds by choosing a *positive* stepsize along the direction determined by two consecutive iterates. That is, given two iterates $w^{k-1}$ and $w^k$, an intermediate point $z^k := w^k + t_k(w^k - w^{k-1})$ for some stepsize $t_k \geq 0$ is first calculated before applying the original algorithmic map ($T^\lambda_{\mathrm{PG}}$ in our case).[2]

---

[2] it is clear that if $t_k \equiv 0$, we reduce back to the original algorithm.

Another popular acceleration scheme for gradient algorithms is the spectral approach pioneered by [5]. They take the differences of the gradients and of the iterates in two consecutive iterations to estimate the curvature at the current point, and use it to decide the step size for updating along the reversed gradient direction. It has been shown in [39] that equipping this step size with a backtracking procedure leads to significantly faster convergence for proximal gradient on regularized optimization problems, which includes our PG for (1) as a special case.

To describe our proposed double acceleration procedure that combines extrapolation and spectral techniques, we first observe that all PG iterates lie on $A_s$, and that $A_s$ can be finitely decomposed as (7). When two consecutive iterates lie on the same convex subspace $A_J$ for some $J \in \mathcal{J}_s$, within these two iterations, we are actually conducting convex optimization. In this case, an extrapolation step within $A_J$ is reasonable because it will not violate the constraint, and acceleration can be expected from the improved rates of accelerated proximal gradient on convex problems in [9, 32]. Judging from Theorem 2.1 (b), the corresponding $J$ is also a candidate index set that belongs to $\mathcal{I}_{w^*}$, so extrapolation within $A_J$ makes further sense. We set $t_k = 0$ to skip the extrapolation step if $d^k$ is not a descent direction for $f$ at $w^k$. Otherwise, we start from some $\hat{t}_k > 0$ decided by the curvature information of $f$, and then execute a backtracking linesearch along $d^k := w^k - w^{k-1}$ to set $t_k = \eta^i \hat{t}_k$ for the smallest integer $i \geq 0$ that provides sufficient descent

$$f(w^k + t_k d^k) \leq f(w^k) - \sigma t_k^2 \|d^k\|^2, \tag{12}$$

given parameters $\eta, \sigma \in (0, 1)$. We then apply (5) to $z^k = w^k + t_k d^k$ to obtain $w^{k+1}$.

For the spectral initialization $\hat{t}_k$ for accelerating the convergence, instead of directly using approaches of [5, 39] that takes the reversed gradient as the update direction, we need to devise a different mechanism as our direction $d^k$ is not directly related to the gradient. We observe that for the stepsize

$$\alpha_k := \langle s^k, s^k \rangle / \langle s^k, r^k \rangle, \quad s^k := w^k - w^{k-1}, \quad r^k := \nabla f(w^k) - \nabla f(w^{k-1}) \tag{13}$$

used in [5], the final update $-\alpha_k \nabla f(w^k)$ is actually the minimizer of the following subproblem

$$\min_{d \in \Re^n} \quad \langle \nabla f(w^k), d \rangle + \|d\|^2 / (2\alpha_k). \tag{14}$$

By juxtaposing the above quadratic problem and the upper bound provided by the descent lemma [7, Lemma 5.7], we can view $\alpha_k^{-1}$ as an estimate of the local Lipschitz parameter that could be much smaller than $L$ but still guarantee descent of the objective. We thus follow this idea to decide $\hat{t}_k$ using such curvature estimate and the descent lemma by

$$\hat{t}_k = \arg\min_{t \geq 0} \ \langle \nabla f(w^k), t d^k \rangle + \|t d^k\|^2 / (2\alpha_k) \quad \Leftrightarrow \quad \hat{t}_k = -\langle \alpha_k \nabla f(w^k), d^k \rangle / \|d^k\|^2. \tag{15}$$

Another interpretation of (13) is that $\alpha_k^{-1} I$ also serves as an estimate of $\nabla^2 f(w^k)$,[3] and the objective in (14) is a low-cost approximation of the second-order Taylor expansion of $f$. However, we notice that for problems in the form of (4) and with $d^k \in A_J$, the exact second-order Taylor expansion

$$f(w^k + t d^k) \approx f(w^k) + t \langle \nabla f(w^k), d^k \rangle + t^2 \langle \nabla^2 f(w^k) d^k, d^k \rangle / 2 \tag{16}$$

can be calculated efficiently. In particular, for (4) and any $d^k \in A_J$, we get from $X d^k = X_{:,J} d_J^k$:

$$\begin{aligned}
\nabla f(w^k)^\top d^k &= \nabla g \left( (X w^k) \right)^\top \left( X_{:,J} d_J^k \right), \\
\langle \nabla^2 f(w^k) d^k, d^k \rangle &= \langle (X_{:,J} d_J^k), \nabla^2 g \left( (X w^k) \right) (X_{:,J} d_J^k) \rangle,
\end{aligned} \tag{17}$$

which can be calculated in $O(ms)$ time by computing $X_{:,J} d_J^k$ first. This $O(ms)$ cost is much cheaper than the $O(mn)$ one for evaluating the full gradient of $f$ needed in the PG step, so our extrapolation plus spectral techniques has only negligible cost. Moreover, for our case of $d^k = w^k - w^{k-1}$, we can further reduce the cost of calculate $X_{:,J} d_J^k$ and thus (17) to $O(m)$ by recycling intermediate computational results needed in evaluating $f(w^k)$ through $X_{:,J} d_J^k = X w^k - X w^{k-1}$. With such tricks for efficient computation, we therefore consider the more accurate approximation to let $\hat{t}_k$ be

---

[3] As $\nabla f$ is Lipschitz continuous, it is differentiable almost everywhere. Here, we denote by $\nabla^2 f(w^k)$ a generalized Hessian of $f$ at $w$, which is well-defined for $f$ with Lipschitz continuous gradient [19].

the scalar that minimizes the quadratic function on the right-hand side of (16) for problems in the form (4). That is, we use

$$\hat{t}_k := - \left\langle \nabla f(w^k), d^k \right\rangle / \left\langle \nabla^2 f(w^k) d^k, d^k \right\rangle. \tag{18}$$

Finally, for both (18) and (15), we safeguard $\hat{t}_k$ by

$$\hat{t}_k \leftarrow P_{[c_k \alpha_{\min}, c_k \alpha_{\max}]} \left( \hat{t}_k \right) \tag{19}$$

for some fixed $\alpha_{\max} \geq \alpha_{\min} > 0$, where

$$c_k := \left\| (\nabla f(w^k))_J \right\| / (\zeta_k \| d^k \|), \quad \zeta_k := - \left\langle d^k, \nabla f(w^k) \right\rangle / (\| d^k \| \| (\nabla f(w^k))_J \|) \in (0, 1]. \tag{20}$$

We also note that the low cost of evaluating $Xd^k$ is also the key to making the backtracking in (12) practical, as each $f(w^k + \eta^i \hat{t}_k d^k)$ can be calculated in $O(m)$ time through linear combinations of $Xw^k$ and $Xd^k$. The above procedure is summarized in Algorithm 1 with global convergence guaranteed by Theorem 3.1. In Theorem 3.2, we establish its full convergence as well as its convergence rates under a KL condition at $w^*$: there exists neighborhood $U \subset \Re^n$ of $w^*$, $\theta \in [0, 1]$, and $\kappa > 0$ such that for every $J \in \mathcal{I}_{w^*}$,

$$(f(w) - f(w^*))^\theta \leq \kappa \| (\nabla f(w))_J \|, \quad \forall w \in A_J \cap U. \tag{21}$$

We denote by $n_k$ the number of successful extrapolation steps in the first $k$ iterations of Algorithm 1. The part of $\theta \in [0, 1/2]$ with $f$ being convex in the last item of Theorem 3.2 is directly from the result of [25].

**Theorem 3.1.** *Under the hypotheses of Theorem 2.1, any accumulation point of a sequence generated by Algorithm 1 is a stationary point.*

**Theorem 3.2.** *Consider either* (5) *or Algorithm 1 with* $\eta, \sigma, \epsilon \in (0, 1)$, *and* $\alpha_{\max} \geq \alpha_{\min} > 0$, *and suppose that there is an accumulation point* $w^*$ *of the iterates at which the KL condition holds. Then* $w^k \to w^*$. *Moreover, the following rates hold:*

*(a) If* $\theta \in (1/2, 1)$*:* $f(w^k) - f(w^*) = O((k + n_k)^{-1/(2\theta - 1)})$.
*(b) If* $\theta \in (0, 1/2]$*:* $f(w^k) - f(w^*) = O(\exp(-(k + n_k)))$.
*(c) If* $\theta = 0$, *or* $\theta \in [0, 1/2]$ *and* $f$ *is convex: there is* $k_0 \geq 0$ *such that* $f(w^k) = f(w^*)$ *for all* $k \geq k_0$.

We stress that convexity of $f$ is not required in Theorems 3.1 and 3.2 except the second half of the last item of Theorem 3.2. There are several advantages of the proposed extrapolation strategy over existing ones in [27, 37]. The most obvious one is the faster rates in Theorem 3.2 over PG such that each successful extrapolation step in our method contributes to the convergence speed, while existing methods only provide the same convergence speed as PG. Next, existing strategies only use prespecified step sizes without information from the given problem nor the current progress, and they only restrict such step sizes to be within $[0, 1]$. Our method, on the other hand, fully takes advantage of the function curvature and can allow for arbitrarily large step sizes to better decrease the objective. In fact, we often observe $t_k \gg 1$ in our numerical experiments. Moreover, our acceleration techniques utilize the nature of (7) and (4) to obtain very efficient implementation for ERM problems such that the per-iteration cost of Algorithm 1 is almost the same as that of PG, while the approach of [27] requires evaluating $f$ and $\nabla f$ at two points per iteration, and thus has twice the per-iteration cost.

A finite termination result similar to Theorem 3.2 (c) is presented in [29] under a Hölderian error bound that is closely related to the KL condition, but their result requires convexity of both the smooth term and the regularizer, so it is not applicable to (1) that involves a nonconvex constraint.

## 3.2 Subspace Identification

In line with the above discussion, we interpret (8) as a theoretical property guaranteeing that the iterates of the projected gradient algorithm (5) will eventually identify the subspaces $A_J$ that contain a candidate solution $w^*$ after a finite number of iterations. Consequently, the task of minimizing $f$ over the nonconvex set $A_s$ can be reduced to a convex optimization problem of minimizing $f$ over $A_J$. Motivated by this, we present a two-stage algorithm described in Algorithm 2 that switches to a high-order method for smooth convex optimization after a candidate piece $A_J$ is identified to

**Algorithm 1:** Accelerated projected gradient algorithm by extrapolation (APG)

---

1  Given an initial vector $w^0 \in \Re^n$ and parameters $\epsilon, \eta, \sigma \in (0,1)$, $\alpha_{\max} \geq \alpha_{\min} > 0$, $\lambda \in (0, 1/L)$.

2  **for** $k = 0, 1, 2, \ldots$ **do**

3     **if** $k > 0$; $w^{k-1}$ and $w^k$ activate the same $A_J$; and $\zeta_k \geq \epsilon$ **then**

4         $d^k \leftarrow w^k - w^{k-1}$, and compute $\hat{t}_k$ from (19) with either (15) or (18)

5         **for** $i = 0, 1, \ldots$ **do**

6             $t_k \leftarrow \eta^i \hat{t}_k$

7             **if** (12) is satisfied **then** $z^k \leftarrow w^k + t_k d^k$, and break

8     **else** $z^k \leftarrow w^k$

9     $w^{k+1} \leftarrow T_{\text{PG}}^\lambda(z^k)$

---

obtain even faster convergence. Since $\nabla f$ is assumed to be Lipschitz continuous, the generalized Hessian of $f$ exists everywhere [19], so we may employ a semismooth Newton (SSN) method [35] with backtracking linesearch to get a faster convergence speed with low cost (details in Appendix A). In particular, we reduce the computation costs by considering the restriction of $f$ on the subspace $A_J$ by treating the coordinates not in $J$ as non-variables so that the problem considered is indeed smooth and convex. As we cannot know a priori whether $I_{w^*}$ is indeed identified, we adopt the approach implemented in [26, 28, 23] to consider it identified when $w^k$ activates the same $A_J$ for long enough consecutive iterations. To further safeguard that we are not optimizing over a wrong subspace, we also incorporate the idea of [38, 4, 28, 23] to periodically alternate to a PG step (5) after switching to the SSN stage. A detailed description of this two-stage algorithm is in Algorithm 2.

In the following theorem, we show that superlinear convergence can be obtained for Algorithm 2 even if we take only one SSN step every time between two steps of (5), using a simplified setting of twice-differentiability. For our next theorem, we need to introduce some additional notations. Given any $w \in A_J$, we use $f_J(w_J) := f(w)$ to denote the function of considering only the coordinates of $w$ in $J$ as variables and treating the remaining as constant zeros. We assume that the conditions of Theorem 2.1 (b) hold with $w^* \in A_s$, and that $f$ is twice-differentiable around a neighborhood $U$ of $w^*$ with $\nabla^2 f_J$ Lipschitz continuous in $U$ and $\nabla^2 f_J(w^*)$ positive definite for all $J \in \mathcal{I}_{w^*}$.

**Theorem 3.3.** *Suppose that starting after $k \geq N$ and $P_{A_s}(w^k) \subset U$, we conduct $t$ Newton steps between every two steps of (5) for $t \geq 1$:*

$$w^{k,0} \in P_{A_s}(w^k), \quad \begin{cases} J & \in \mathcal{I}_{w^{k,0}}, \\ w_i^{k,j+1} & = 0, \quad \forall i \notin J, \quad j = 1, \ldots, t-1, \quad w^{k+1} \in T_{\text{PG}}^\lambda(w^{k,t}). \\ w_J^{k,j+1} & = w_J^{k,j} - \nabla^2 f_J(w_J^{k,j})^{-1} \nabla f_J(w_J^{k,j}), \end{cases} \quad (22)$$

*Then $w^k \to w^*$ at a Q-quadratic rate.*

In practice, the linear system for obtaining the SSN step is only solved inexactly via a (preconditioned) conjugate gradient (PCG) method, and with suitable stopping conditions for PCG and proper algorithmic modifications such as those in [40, 30], superlinear convergence can still be obtained easily. Interested readers are referred to Appendix A for a more detailed description of our implementation.

## 4  Experiments

In this section, we conduct numerical experiments to demonstrate the accelerated techniques presented in Section 3. We employ Algorithm 1 (APG) with (18) to accelerate PG, and further accelerate APG by incorporating subspace identification described in Algorithm 2, which we denote by APG+.[4] Comparisons with the extrapolated PG algorithm of Li and Lin [27], which we denote by PG-LL, are also presented. PG-LL is a state-of-the-art approach for nonconvex regularized optimization and thus suitable for (1). For $f$ in (1), we consider both LS (3) and logistic regression (LR)

$$f(w) = \sum_{i=1}^m \log \left( 1 + \exp \left( -y_i x_i^\top w \right) \right) + \mu \|w\|^2 / 2, \quad (23)$$

---

[4] That is, if $Unchanged < S$ in Algorithm 2, we calculate $z^k$ as in Algorithm 1

**Algorithm 2:** Accelerated projected gradient algorithm by subspace identification (PG+)

**1** Given an initial vector $w^0 \in \Re^n$ and $S, t \in \mathbb{N}$. Set Unchanged $\leftarrow 0$.
**2** **for** $k = 0, 1, 2, \ldots$ **do**
**3**     **if** $k > 0$*, and $w^{k-1}$ and $w^k$ activate the same component of $A_s$* **then**
**4**        Let $J \in \mathcal{J}_s$ correspond to the activated component
**5**        Unchanged $\leftarrow$ Unchanged $+1$
**6**     **else** Unchanged $\leftarrow 0$
**7**     **if** *Unchanged* $\geq S$ **then**
**8**        $y^k \leftarrow P_{A_J}(w^k)$ and use $t$ steps of SSN described in Appendix A, starting from $y^k$, to find $z^k$ that approximately minimizes $f|_{A_J}$
**9**        **if** *SSN fails* **then** $z^k \leftarrow w^k$ and Unchanged $\leftarrow 0$.
**10**     **else** $z^k \leftarrow w^k$
**11**     $w^{k+1} \leftarrow T_{\mathrm{PG}}^\lambda(z^k)$

where $(x_i, y_i) \in \Re^n \times \{-1, 1\}$, $i = 1, \ldots, m$, are the training instances, and $\mu > 0$ is a small regularization parameter added to make the logistic loss coercive.

The algorithms are implemented in MATLAB and tested with public datasets in Tables 2 and 3 in Appendix B. All algorithms compared start from $w^0 = 0$ and terminate when the first-order optimality condition

$$\text{Residual}(w) := \|w - P_{A_s}(w - \lambda \nabla f(w))\|/(1 + \|w\| + \lambda\|\nabla f(w)\|) < \hat{\epsilon} \qquad (24)$$

is met for some given $\hat{\epsilon} > 0$. More setting and parameter details of our experiments are in Appendix B.

**Comparisons of algorithms for large datasets.** To fit the practical scenario of using (1), we specifically selected high-dimensional datasets with $n$ larger than $m$. We conduct experiments with various $s$ to widely test the performance under different scenarios. In particular, we consider $s \in \{\lceil 0.01m \rceil, \lceil 0.05m \rceil, \lceil 0.1m \rceil\}$ on all data except for the largest dataset webspam, for which we set $s \in \{\lceil 0.001m \rceil, \lceil 0.005m \rceil, \lceil 0.01m \rceil\}$. The results of the experiment with the smallest $s$ are summarized in Figure 1, and results of the other two settings of $s$ are in Appendix C.

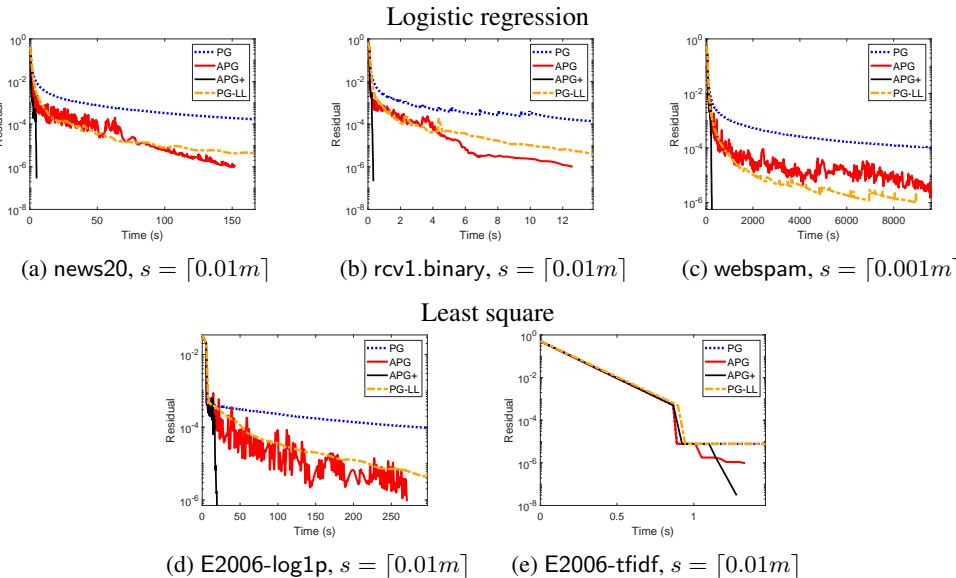

Figure 1: Experiment on sparse regularized LR and LS. We present time v.s. residual in (24).

Evidently, the extrapolation procedure in APG provides a significant improvement in the running time compared with the base algorithm PG, and further incorporating subspace identification as in APG+ results to a very fast algorithm that outperforms PG and APG by magnitudes. Since the per-iteration

Table 1: Comparison of algorithms for (1) to meet (24) with $\hat{\epsilon} = 10^{-6}$, with (23) and (3) and with sparsity levels $s_1 = \lceil 0.01m \rceil$ and $s_2 = \lceil 0.05m \rceil$ for all datasets except webspam where $s_1 = \lceil 0.001m \rceil$ and $s_2 = \lceil 0.005m \rceil$. CPU: CPU time in seconds. GE: number of gradient evaluations. In one iteration, PG, APG, and APG+ needs one gradient evaluation , while PG-LL and PG-LL+ needs two. CG: number of Hessian-vector products in the PCG procedure for obtaining SSN steps. PA: prediction accuracy (for (23)). MSE: mean-squared error (for (3)). Time with $*$ indicates that the algorithm is terminated after running 10000 iterations without satisfying (24).

| Dataset | Method | $s_1$ | | | | $s_2$ | | | |
|---|---|---|---|---|---|---|---|---|---|
| | | CPU | GE | CG | PA | CPU | GE | CG | PA |
| news20 | PG | *738.7 | 10000 | 0 | 0.877 | *728.9 | 10000 | 0 | 0.935 |
| | APG | 151.7 | 1583 | 0 | 0.877 | 758.3 | 8428 | 0 | 0.923 |
| | APG+ | **5.0** | 52 | 63 | 0.853 | **16.1** | 171 | 67 | 0.923 |
| | PG-LL | 366.7 | 4682 | 0 | 0.873 | *1494.4 | 20000 | 0 | 0.922 |
| | APG-LL+ | 6.6 | 152 | 88 | 0.854 | 29.2 | 417 | 89 | 0.920 |
| rcv1.binary | PG | *58.4 | 10000 | 0 | 0.937 | *72.7 | 10000 | 0 | 0.951 |
| | APG | 12.6 | 1120 | 0 | 0.935 | 82.4 | 6372 | 0 | 0.934 |
| | APG+ | **0.3** | 21 | 42 | 0.931 | **2.4** | 192 | 138 | 0.940 |
| | PG-LL | 22.2 | 3638 | 0 | 0.935 | 72.1 | 8738 | 0 | 0.929 |
| | APG-LL+ | 0.6 | 99 | 49 | 0.930 | 4.9 | 626 | 236 | 0.939 |
| webspam | PG | *18660.1 | 10000 | 0 | 0.964 | *30776.2 | 10000 | 0 | 0.978 |
| | APG | 19683.4 | 7682 | 0 | 0.981 | 7722.4 | 2008 | 0 | 0.991 |
| | APG+ | **248.3** | 75 | 88 | 0.969 | **695.4** | 164 | 57 | 0.991 |
| | PG-LL | 9001.3 | 4720 | 0 | 0.972 | 10163.5 | 3098 | 0 | 0.990 |
| | APG-LL+ | 447.3 | 264 | 92 | 0.965 | 837.3 | 294 | 90 | 0.992 |
| | | CPU | GE | CG | MSE | CPU | GE | CG | MSE |
| E2006-log1p | PG | *2998.6 | 10000 | 0 | 0.167 | *3644.1 | 10000 | 0 | 0.161 |
| | APG | 270.6 | 669 | 0 | 0.136 | 811.8 | 1757 | 0 | 0.133 |
| | APG+ | **19.5** | 40 | 49 | 0.141 | **105.6** | 222 | 124 | 0.132 |
| | PG-LL | *6049.8 | 20000 | 0 | 0.132 | 2696.0 | 7086 | 0 | 0.132 |
| | APG-LL+ | 41.2 | 142 | 38 | 0.142 | 107.5 | 326 | 100 | 0.138 |
| E2006-tfidf | PG | *242.7 | 10000 | 0 | 0.152 | *666.9 | 10000 | 0 | 0.152 |
| | APG | **1.3** | 14 | 0 | 0.154 | **3.3** | 33 | 0 | 0.153 |
| | APG+ | **1.3** | 8 | 6 | 0.141 | **3.3** | 31 | 7 | 0.139 |
| | PG-LL | 110.6 | 4440 | 0 | 0.152 | 304.8 | 4558 | 0 | 0.151 |
| | APG-LL+ | 1.7 | 34 | 6 | 0.141 | 3.7 | 47 | 7 | 0.139 |

cost of PG and APG are almost the same as argued in Section 3, we note that the convergence of APG in terms of iterations is also superior to that of PG.

We also report the required time and number of gradient evaluations (which is the main computation at each iteration) for the algorithms to drive (24) below $\hat{\epsilon} = 10^{-6}$. For PG, APG, and APG+, one gradient evaluation is needed per iteration, so the number of gradient evaluations is equivalent to the iteration count. For PG-LL, two gradient evaluations are needed per iteration, so its cost is twice of other methods. We also report the prediction performance on the test data, and we in particular use the test accuracy for (23) and the mean-squared error for (3). Results for the two smaller $s$ are in Table 1 while that for the largest $s$ is in Appendix C. It is clear from the results in Table 1 that APG outperforms PG-LL for most of the test instances considered, while APG+ is magnitudes faster than PG-LL. When we equip PG-LL with our acceleration techniques by replacing $T_{\mathrm{PG}}^{\lambda}$ in Algorithms 1 and 2 with the algorithmic map defining PG-LL, we can further speed up PG-LL greatly as shown under the name APG-LL+ (see Table 1). We do not observe a method that consistently possesses the best prediction performance, as this is mainly affected by which local optima is found, while no algorithm is able to find the best local optima among all candidates. With no prediction performance degradation, we see that APG+ and APG-LL+ reduce the time needed to solve (1) to a level significantly lower than that of the state of the art.

In Appendix C.3, we demonstrate the effect on prediction performance when we vary the residual (24) and illustrate that tight residual level is indeed required to obtain better prediction. Comparisons with a greedy method is shown in Appendix C.4.

**Transition Plots.** To demonstrate the behavior of the algorithm for increasing values of $s$, we fit the smaller datasets in Table 3 using logistic loss (23) and least squares loss (3) for varying $s = \lceil km \rceil$, where $k = 0.2, 0.4, 0.6, \ldots, 3$. The transition plots are presented in Figure 2. We note that the time is in log scale.

We can see clearly that APG+ and APG-LL+ are consistently magnitudes faster than the baseline PG method throughout all sparsity levels. On the other hand, the same-subspace extrapolation scheme of APG is consistently faster than PG and APG-LL and slower than the two Newton acceleration schemes, although the performance is sometimes closer to APG+/APG-LL+ while sometimes closer to PG. APG-LL tends to outperform PG in most situations as well, but in several cases when solving the least square problem, especially when $s$ is small, it can sometimes be slower than PG. Overall speaking, the results in the transition plots show that our proposed acceleration schemes are indeed effective for all sparsity levels tested.

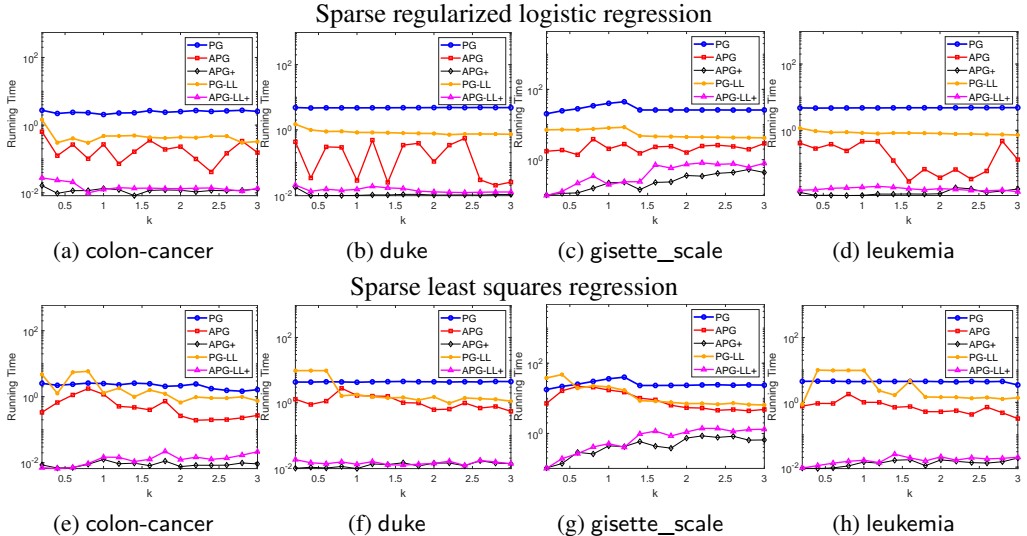

Figure 2: Transition plots. We present sparsity levels versus running time (in log scale). Top row: logistic loss. Bottom row: least square loss.

## 5 Conclusions

In this work, we revisited the projected gradient algorithm for solving $\ell_0$-norm constrained optimization problems. Through a natural decomposition of the constraint set into subspaces and the proven ability of the projected gradient method to identify a subspace that contains a solution, we further proposed effective acceleration schemes with provable convergence speed improvements. Experiments showed that our acceleration strategies improve significantly both the convergence speed and the running time of the original projected gradient algorithm, and outperform the state of the art for $\ell_0$-norm constrained problems by a huge margin. We plan to extend our analysis and algorithm to the setting of a nonconvex objective in the near future.

## Acknowledgments

This work was supported in part by Academia Sinica Grand Challenge Program Seed Grant No. AS-GCS-111-M05 and NSTC of R.O.C. grants 109-2222-E-001-003 and 111-2628-E-001-003.

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
