# OpenReview forum: "Accelerated Projected Gradient Algorithms for Sparsity Constrained Optimization Problems"
_NeurIPS.cc/2022/Conference — NeurIPS 2022 Accept_

### Official Review · Reviewer_TQ21 · 2022-06-16

**Rating:** 7
**Confidence:** 4
**Soundness:** 3 good
**Presentation:** 3 good
**Contribution:** 3 good

**Summary:**

The authors study projected gradient descent for minimizing a smooth convex function under a sparsity constraint. They provide acceleration strategies

**Questions:**

I am wondering about the only assumption that f is lower bounded. This is not sufficient for a solution to exist, for instance log(1+exp(-x)) is lower bounded but does not have a minimizer. So, in Theorem 2.1, for instance, an accumulation point does not necessarily exist, and in that case the statements say nothing. It would be good to give sufficient conditions on f such that an accumulation point exists.

A minor mistake: in Theorem 2.1 (a): either f(w^k) is strictly decreasing, or it becomes constant, i.e. w^{k+1}=w^k, which means the algorithm is exactly at a critical point.

typo: eq. 8, replace the period at the end by a comma



**Limitations:**

I don't see any limitation.

**Strengths And Weaknesses:**

Strengths: although projected gradient descent with this nonconvex constraint set is an old algorithm, the findings in this paper are new, to the best of my knowledge. I have checked some of the proofs and they seem correct to me. The experiments show that the acceleration strategies work very well in practice.

Weakness: this is minor, but some relevant papers are not cited. For instance the algorithm is often called iterative hard-thresholding and classical papers on the topic are "Iterative hard thresholding for compressed sensing", Blumensath and Davies, 2009,
"Accelerated iterative hard thresholding", Blumensath, 2012.

---

> ### Author Response · Authors · 2022-08-02
> **Response to TQ21**
>
> > this is minor, but some relevant papers are not cited. For instance the algorithm is often called iterative hard-thresholding and classical papers on the topic are "Iterative hard thresholding for compressed sensing", Blumensath and Davies, 2009, "Accelerated iterative hard thresholding", Blumensath, 2012.
>
> These references are added to our revised manuscript; please see [11,12].
>
> > I am wondering about the only assumption that $f$ is lower bounded. This is not sufficient for a solution to exist, for instance $\log(1+\exp(-x))$ is lower bounded but does not have a minimizer. So, in Theorem 2.1, for instance, an accumulation point does not necessarily exist, and in that case the statements say nothing. It would be good to give sufficient conditions on $f$ such that an accumulation point exists.
>
> Thank you for this question. Lower-boundedness of $f$ is primarily needed to guarantee that the decreasing sequence $\{f(w^k)\}$ will converge to a finite value (see equation (31) in Appendix D.1). As mentioned by the reviewer, lower-boundedness is indeed not sufficient to guarantee the existence of a solution nor of accumulation points in Theorem 2.1.  We highlight that we have indicated in Lines 129-130 that a sufficient condition for the existence of accumulation points is the coercivity of $f$ over each linear subspace $A_J$.
>
> > A minor mistake: in Theorem 2.1 (a): either $f(w^k)$ is strictly decreasing, or it becomes constant, i.e. $w^{k+1}=w^k$, which means the algorithm is exactly at a critical point.
>
> Thanks for this correction. We have updated Theorem 2.1(a) and its proof accordingly in the revised manuscript.

---

### Official Review · Reviewer_Kgs5 · 2022-07-10

**Rating:** 7
**Confidence:** 4
**Soundness:** 4 excellent
**Presentation:** 3 good
**Contribution:** 3 good

**Summary:**

This paper presents a suite of projected gradient algorithms for minimizing convex L-Lipschitz functions over the non-convex $\ell_0$ norm-bounded set. Specifically, the authors theoretically show that projected gradient descent converges at a locally linear rate to a stationary point of the objective function. Moreover, under certain  assumptions, they propose two different acceleration schemes based on a) an extrapolation  combined with spectral initialization for the step-size and b) semismooth Newton updates after correctly indetifying the subspace, that lead to improved rates of convergence over the standard PGD method. Finally, the merits of the proposed algorithms are shown on real data experimental results.

---------------------------------------------
Post-rebuttal update: The authors have addressed my comments and concerns in their rebuttal. Thus, I raise my score to Accept.


**Questions:**

1) The authors provide the convergence rate for the iterates of PGD in Section 2, then the rate of convergence for the objective for the accelerated using extrapolation algorithms, and finally, the quadratic convergence rate for the iterates of the semi-smooth Newton-type algorithm. Could the authors  also provide convergence rates for the updates of APGD (i.e.,  the extrapolation-based approach).
2) Is there any advantage by considering a general projection operation? Wouldn't be easier (in terms of the uniqueness of the projections)  if you had restricted your attention to orthogonal projections ?
3) @Theorem 2.1:  It is not clear how $w^\ast$ is defined when $T^{\lambda}_{PG}(w^\ast)$ is not singleton. Could it be any element of the returned set? Can the authors elaborate more on this?

**Limitations:**

The authors adequately addressed the limitations and potential negative societal impact of their work.

**Strengths And Weaknesses:**

Strengths:
1) The paper is easy to follow and the authors succeed in conveying the main ideas to the readers.
2) The theoretical contributions of the paper seem to be interesting. The authors show an improved R-linear rate of convergence of the objective function when using the extrapolation step.  Moreover, a quadratic convergence rate for the updates is attained when the proposed semismooth Newton type algorithm is used.
3) The theoretical results improve on previously known state-of-art results relaxing the required assumptions and covering more general classed of objective functions.
4) The experimental results on various real-world datasets demonstrate the efficiency of the proposed algorithms validating the theory provided.

Weaknesses:
1) The introduction could be improved using subsections for the related work and contributions.
2)  The statements in the theorems are not always clearly written. Namely, the assumptions used in the theorems are stated within the theorem making it hard to follow the final conclusion. I suggest the authors state the assumptions before the theorems.
3) The authors do not elaborate on why the assumptions used in their theorems are weaker than the ones based on KL condition. They should better explain those points and extend the paragraph in lines 127-131.
4) The authors do not sufficiently motivate the use of $\ell_0$ norm over convex $\ell_1$ or other non-smooth and non-convex regularizers.
   In the experimental results section, the authors show that the PG-LL can converge significantly faster when the acceleration schemes proposed in the paper are adopted. That is an interesting observation but also raises the question: When the $\ell_0$ norm constraint should be preferred over other convex or nonsmooth nonconvex constraints:
5) The authors do not provide phase transition plots to show the limitation of their approach as the sparsity level $s$ increases. Also, comparison with the algorithm of [9], which is an alternative approach for the same problem is not provided.

---

> ### Author Response · Authors · 2022-08-02
> **Response to Kgs5 (Part 1 of 2)**
>
> > The introduction could be improved using subsections for the related work and contributions.
>
> Thank you for this suggestion. Paragraph titles are now added in the introduction section of our revised manuscript (we decided not to use numbered subsections for the interest of page length, but would probably be able to switch to subsections in the camera-ready version).
>
> > The statements in the theorems are not always clearly written. Namely, the assumptions used in the theorems are stated within the theorem making it hard to follow the final conclusion. I suggest the authors state the assumptions before the theorems.
>
> Thank you for this helpful suggestion. In view of this, we have revised the statements in Theorems 3.2 and 3.3 to make it easier to follow the conclusions. As for Theorem 2.1, its statements (a), (b), and (c) have different hypotheses, so stating the assumptions before the theorem itself may not be ideal. We hope that these changes are satisfactory to the reviewer.
>
> > The authors do not elaborate on why the assumptions used in their theorems are weaker than the ones based on KL condition. They should better explain those points and extend the paragraph in lines 127-131.
>
> Theorem 2.1(b) does not require any regularity assumption at all, and is therefore clearly weaker than adding any hypothesis such as the KL condition.
> In particular, the KL condition requires the objective value to decrease with the minimum-norm subgradient in a neighborhood of the accumulation point, but our result only requires a very mild condition at the accumulation point itself but has no restriction around its neighborhood. (We have also explained in Lines 123-126 about why this condition is very mild and almost always true in practice.)
> In the revision, we now have added some brief explanations about the reasoning above in Lines 133-135.
>
> On the other hand, fulfillment of the hypothesis of Theorem 2.1(c) merely requires strong convexity of $f$ over the subspaces $A_J$ containing $w^*$, which we have mentioned in Lines 126-127. This is easily verifiable compared to the KL condition (21) that requires determining the appropriate neighborhood $U$ of $w^{*}$ as well as positive constants $\theta$ and $\kappa$. Due to space limitations, we only added a brief pointer in Line 133, before stating Theorem 2.2, to emphasize the KL condition.
>
> > The authors do not sufficiently motivate the use of  $\ell_0$ norm over convex  $\ell_1$ or other non-smooth and non-convex regularizers.
> In the experimental results section, the authors show that the PG-LL can converge significantly faster when the acceleration schemes proposed in the paper are adopted. That is an interesting observation but also raises the question: When the  $\ell_0$  norm constraint should be preferred over other convex or nonsmooth nonconvex constraints:
>
> We believe that the benefit of using the $\ell_0$ norm over convex surrogates such as the $\ell_1$ norm is well-documented in the literature, and unfortunately, such a rigorous review may not be ideal in the current version of the paper due to space limitations. To make up for this, we refer the readers to the excellent overview provided by Bertsimas et al. [10] in their introduction. Please see Line 42 of the revised paper.
>
> However, a full answer for motivating the usage of the $\ell_0$ norm over other regularizers or when will the former be better than the latter should be application-dependent and also more relevant to their statistical properties, which are beyond the scope of the work, and we believe this could itself be the whole content of another paper(s).
> Our goal in this work is not to advocate that one should always consider the $\ell_0$ norm constraint. Instead, we aim at removing one existing disadvantage of this formulation: the possibly lengthier training time. The major purpose of our work is to greatly improve the efficiency of solving $\ell_0$-norm constrained problems, and thus when people consider about which formulation to use, they will not count in the efficiency in solving this problem as a factor for consideration.

---

> ### Author Response · Authors · 2022-08-02
> **Response to Kgs5 (Part 2 of 2)**
>
> > The authors do not provide phase transition plots to show the limitation of their approach as the sparsity level  $s$ increases. Also, comparison with the algorithm of [9], which is an alternative approach for the same problem is not provided.
>
> Thanks for this valuable suggestion that has strengthened our numerical result to further demonstrate the efficiency of the proposed acceleration schemes.
> We have now added transition plots in Appendix C.4. The results clearly show that the two Newton-accelerated approaches APG+ and APG-LL+ are consistently around two magnitudes faster than PG in all sparsity levels we tested, and APG is also steadily faster than APG-LL and PG but slower than APG+ and APG-LL+.
> On the other hand, PG-LL is sometimes slower than the base PG algorithm especially when $s$ is small, but in other situations it tends to outperform PG.
>
> In [9] (now [10] in the revised manuscript, and we will use [10] to refer this work from here on), the authors have already compared their MIP algorithm with the baseline PG method. Their observation is that MIP alone is even slower than PG, but MIP can be significantly accelerated by using PG as a warmstart mechanism. But we note that our methods can also be combined with MIP in the same fashion. And as our methods are faster than PG for generating a warmstart point, the combination with MIP of our methods will also certainly be faster than the combination of PG and MIP in [10].
>
> > The authors provide the convergence rate for the iterates of PGD in Section 2, then the rate of convergence for the objective for the accelerated using extrapolation algorithms, and finally, the quadratic convergence rate for the iterates of the semi-smooth Newton-type algorithm. Could the authors also provide convergence rates for the updates of APGD (i.e., the extrapolation-based approach).
>
> Could you clarify about convergence rates of APGD under *what circumstances* you are looking for? As you have already mentioned that we have provided "rate of convergence for the objective for the accelerated using extrapolation algorithms," which seems to us exactly the answer to your question.
> If you meant the case without KL conditions, unfortunately we do not have such a result because our analysis in Theorem 2.2 relies on the iterates behavior of PG and is not applicable when extrapolations intertwine with PG iterations.
> But a good approximation for that is to let $\theta \rightarrow 1$ (for the convex case) or to consider the case of $\theta = 1/2$ (for the strongly convex case) in Theorem 3.2.
> In particular, convexity of $f$ implies that our version of KL condition holds with $\theta = 1$, and strong convexity is sufficient for $\theta = 1/2$.
> And indeed, if no extrapolation is successful (so the algorithm is without extrapolation), those two rates in Theorem 3.2 coincide with what we have for PG in Theorem 2.2.
>
> > Is there any advantage by considering a general projection operation? Wouldn't be easier (in terms of the uniqueness of the projections) if you had restricted your attention to orthogonal projections ?
>
> We can only consider the projection onto $A_s$ due to the nature/definition of the projected gradient approach. In addition, since $A_s$ is nonconvex, projection onto this set is not orthogonal. Hence, unfortunately, we cannot restrict it to orthogonal projections.
>
> One example is when for a point $x$, there are multiple coordinates whose absolute value is the $s$-th largest, then the projection is not unique, and we are unable to get a unique projection. A toy example would be $P_{A_1}(x)$ with $x = (1,1,1)$, then there is no way to decide which sparsity pattern should be selected in this projection. Although indeed we might barely face this situation in practice, for the algorithm to be well-defined, we still need to deal with such degenerate cases.
>
> > @Theorem 2.1: It is not clear how  $w^*$ is defined when $T_{PG}^{\lambda}(w^{*})$ is not singleton. Could it be any element of the returned set? Can the authors elaborate more on this?
>
> The definition is exactly the same as that in equation (5).
> To illustrate this, consider for instance when $n=2$ and $f(w) = 0.5(w_2-1)^2$. In this case, $L=1, \nabla f(w) = (0, w_2-1)$. Let $\lambda$ be any number in $(0,1/L) = (0,1)$, and take the point $w^*= (\lambda, 0)$. By direct calculations, we obtain $T_{PG}^{\lambda}(w^*) = P_{A_s}(\lambda,\lambda) = \{ (\lambda,0) , (0, \lambda)\}$. In other words, $w^*$ belongs to $T_{PG}^{\lambda}(w^*)$ but the set $T_{PG}^{\lambda}(w^*)$ is not a singleton.

---

### Official Review · Reviewer_eyvq · 2022-07-11

**Rating:** 6
**Confidence:** 4
**Soundness:** 3 good
**Presentation:** 3 good
**Contribution:** 3 good

**Summary:**

This paper studied the minimization problem of a convex loss function with an $\ell_0$ constraint. This is a well-known and important subset-selection problem. The authors established the convergence guarantees of the projected gradient algorithm. Two acceleration techniques were further developed with convergence guarantees. Comprehensive numerical studies were carried out to show the favorable acceleration performance.


**Questions:**

From Theorem 3.3 and from observing Figure.1, an interesting question is if one can characterize the point where the quadratic convergence of Newton method kicks in.


**Limitations:**

Given that the major contribution of this paper is theoretical, I think it might be helpful / intreresting to construct a simulated dataset (with known truth) so that one can keep track of the whole algorithm and demonstrate the subspace identification property.


**Strengths And Weaknesses:**

Strengths:

1. Theorem 2.1 established the linear rate of convergence (to a local optimum), which improves upon the sublinear rate an MIP-based approach by Bertsimas et al (2016).

2. Under additional assumptions, the rest of Theorem 2.1 established the convergence characterization under mild assumptions.

3. Two acceleration schemes were proposed, which fully utilize the structure of the $\ell_0$ constraint set. Both schemes are intuitive and easy to implement, and practical considerations and discussions were also provided.

Weakness:

My primary concern is the contraction assumption made for part (c) of Theorem 2.1. This assumption requires that the gradient step is a contraction over all spanned spaces $A_J$ in the decomposition (7) at the accumulation point. It seems unclear how stringent this assumption is. I believe the main takeaway is that certain strong assumption (e.g., strong convexity) does not have to be satisfied for $f$ everywhere, but only for $f$ constrained on $A_J$ for ALL J. However, verifying the contraction property for ALL J seem to be extremely hard (e.g., is it verifiable in polynomial time?). Otherwise, from a practical perspective, requiring global property of $f$ is not really too much different from requiring local (on $A_J$ for all J) property of f.

---

> ### Author Response · Authors · 2022-08-02
> **Response to eyvq**
>
> > My primary concern is the contraction assumption made for part (c) of Theorem 2.1. This assumption requires that the gradient step is a contraction over all spanned spaces $A_J$ in the decomposition (7) at the accumulation point. It seems unclear how stringent this assumption is. I believe the main takeaway is that certain strong assumption (e.g., strong convexity) does not have to be satisfied for $f$ everywhere, but only for  $f$ constrained on $A_J$ for ALL $J$. However, verifying the contraction property for ALL $J$ seem to be extremely hard (e.g., is it verifiable in polynomial time?). Otherwise, from a practical perspective, requiring global property of $f$ is not really too much different from requiring local (on  $A_J$ for all $J$) property of $f$.
>
> Thank you for this comment. Indeed, the interest in requiring the assumption on $f$ to hold only for $A_J$ containing $w^{*}$ is from a theoretical perspective, which is inspired by the restricted isometry property (RIP) usually considered in compressed sensing that is well studied in the literature (for instance, in [12]). To generalize this idea, we have made use of a restricted property of strong convexity to establish the theoretical result, although indeed, this condition is difficult to verify practically just like the RIP condition. Nevertheless, our results give a generalization of what is usually considered in compressed sensing, and therefore we believe that such an extension is meaningful.
>
> Moreover, from the practical point of view, our algorithm is always the same no matter this condition holds or not, and therefore, knowledge of whether this condition holds or not actually does not affect anything in practice. (Running time prediction is always difficult in practice because that is heavily dependent on the dataset.)
>
> > From Theorem 3.3 and from observing Figure.1, an interesting question is if one can characterize the point where the quadratic convergence of Newton method kicks in.
>
> This is a very interesting question that we have also tried to consider in this paper. However, this problem itself is a very difficult problem that might require more sophisticated machinery, and thus, we leave this as our direction for future research.
> In general, the characterization requires knowledge of the value of $|w^*_{\tau(s)}| - |w^*_{\tau(s+1)}|$ (which decides the value of $\delta$ in our proof for Theorem 2.1(b)) for identification.
> After the identification, as in the case of analyzing the quadratic convergence phase of Newton's method like that in Boyd & & Vandenberghe (2004) or Wright & Recht (2022), the point at which quadratic convergence kicks in will depend on the Lipschitz continuity parameter of the Hessian as well as the smallest eigenvalue of $\nabla^2 f_J(w^{*})$.
> But in both practice and theory, there is no way to know these values a priori.
>
> > Given that the major contribution of this paper is theoretical, I think it might be helpful / intreresting to construct a simulated dataset (with known truth) so that one can keep track of the whole algorithm and demonstrate the subspace identification property.
>
> As the problem we consider is essentially a nonconvex one due to the constraint, there is actually no existing algorithm that is guaranteed to converge to the global optimum (i.e., the ground truth). Therefore, it is hard to track even in a simulated data about the correct identification, because it is possible that the iterates actually converge to a different accumulation point that is not the known truth. Ensuring that the iterates indeed converge to the known truth for this demonstrative purpose requires more complicated experimental design, so we plan to conduct it in the near future.
>
> ## References:
> Stephen Boyd and Lieven Vandenberghe. Convex optimization. Cambridge university press, 2004.
>
> Stephen J. Wright and Benjamin Recht. Optimization for data analysis. Cambridge University Press, 2022.

---

> ### Author Response · Authors · 2022-08-07
> **Response to eyvq (update)**
>
> > Given that the major contribution of this paper is theoretical, I think it might be helpful / intreresting to construct a simulated dataset (with known truth) so that one can keep track of the whole algorithm and demonstrate the subspace identification property.
>
> We now have added an experiment in Appendix C.7 to demonstrate the subspace identification property. In contrast to our previous expectation, PG in this case indeed converges to the ground truth, and we have indeed shown that the property holds in practice.

---

> > ### Comment · Reviewer_eyvq · 2022-08-07
> > **Response to the authors**
> >
> > I appreciate the authors' time and effort in addressing my comments.
> >
> > 1. On the conditions:
> >
> > I agree that assumption is hard to verify in practice. I found the statement
> >
> > > ... from the practical point of view, our algorithm is always the same no matter this condition holds or not, and therefore, knowledge of whether this condition holds or not actually does not affect anything in practice.
> >
> > a little confusing. Would it it be possible to construct a setting where the conditions are violated and thus the claimed property no longer holds? Are the authors claiming that the required conditions are actually not tight, thus could be avoided in most cases? Or it is almost impossible to construct a case where such conditions are violated?
> >
> > 2. On the added experiment
> >
> > I thank the authors for taking extra steps in providing these interesting results.

---

> > > ### Author Response · Authors · 2022-08-07
> > > **Conditions clarification**
> > >
> > > What we meant to express is that how the algorithm is executed by a user is irrelevant to whether the conditions hold or not.
> > > It is indeed possible that none of the faster rates in thm 2.1 (c) and all later theorems holds, and only the result of thm 2.1(a) is applicable. But no matter such a condition holds or not, the algorithm itself does not rely on this information to change any parameters.
> > > We were just showing that in certain nicer situations that often take place in practice, we could observe faster rates.
> > > This is what we meant by that the algorithm is always the same (in terms of how it is implemented under various conditions).
> > > And in practice, anyway the algorithm is terminated when a certain stopping condition on the residual (or maybe the objective change) is satisfied but not when a certain number of iterations are executed. Therefore in practice, whether those conditions hold does not really matter for a user.
> > >
> > > On the other hand, one could surely construct a pathelogical case such that the singleton condition is violated.
> > > This condition is violated when the s-th largest element of $|w^* - \lambda \nabla f(w^*)|$ and the (s+1)-th largest one have the same value, as described in L126.
> > > Consider a simplified setting such that each element of $w^* - \lambda \nabla f(w^*)$ follows iid from a certain continuous distribution, then the probability that the s-th and the (s+1)-th largest elements (in absolute value) are of the same value is of probability 0, and thus in this case, the needed condition holds almost surely.
> > >
> > > On the other hand, surely one can easily construct problems that violate the contraction property in thm 2.1c: the simplest case would be setting s=n and considering the least square problem with n>m.
> > > Then the given map will have the smallest eigenvalue being 0 and is not a contraction.
> > > However, as we argued above, no matter this holds or not, the algorithm needs not to know whether this holds. Therefore, again from the practical perspective, we do not need to verify this condition in advance.
> > > Using the same example, it is clear that if we decrease s to be much smaller than n, then with higher and higher chances, our condition will hold, but the suggested global condition will anyway fail. We think that is the main contribution of thm 2.1c in comparison to the suggested global condition.

---

### Official Review · Reviewer_vhqu · 2022-07-12

**Rating:** 6
**Confidence:** 3
**Soundness:** 3 good
**Presentation:** 3 good
**Contribution:** 3 good

**Summary:**

The paper considers $\ell_0$ sparsity constrained minimization problem of a function $f(w)$. It proposes accelerated projected gradient to solve such a problem. The designed acceleration relies on a extrapolation step between two consecutive iterates $w^k$ and $w^{k-1}$ followed by a projection onto the sparsity set, provided that $w^k$ and $w^{k-1}$  lie on the same sparsity subset. The used step-size in the extrapolation is set by backtracking starting from an initial value that is set based on the curvature estimation of $f$. A dedicated algorithm is proposed based on the acceleration scheme. It is supported by theoretical rates of convergence. A second algorithm is sketched ans relies on the fact that if  the support of $w$ is identified the optimization scheme can be carried out using faster Newton-like method by exploiting the Lipschitz continuity of $\nabla f$. The second algorithm periodically alternates between projected gradient and newton-like update. The quadratic rate of convergence is established for the latter algorithm. Empirical evaluation on high dimensional datasets illustrate the effectiveness of the approach.

**Questions:**

* In algorithm 1, how often $w^k$ and $w^{k-1}$  may not activate the same support?
* In practice, how $\alpha_\min$ and $\alpha_\max$ are set in Algorithm 1?
* What are the typical values of $S$ in Algorithm 2? How $S$ influences convergence? Activating the same sparsity support for a certain number $S$ of iterations is not a certificate that the "correct" support is identified nor a convergence in the support is attained.
* How the PG methods perform compared to other existing approaches to deal with $\ell_0$ sparsity problem (such as MIP? greedy methods...)?
* As the prediction performances in Table are almost similar for all methods but the methods differ in computation time, do the same observations hold if one relaxes the stopping criterion $\epsilon$?

**Limitations:**

The paper  discusses the broader impact of the proposed approaches. On my view point the work does not raise any specific issues.


**Strengths And Weaknesses:**

* Overall, the paper is well written and  well  organized. The rationale behind the proposed accelerated projected gradient is justified and the new novelty brought by  the approach is stated clearly.

* A main contribution of the main paper is the proposition of an acceleration, based on extrapolation technique, within a projected gradient scheme for $\ell_0$ sparsity optimization. The feature of the approach is to extrapolate between two consecutive iterates with the same sparsity support. Complemented with a linesearch scheme with appropriate initial estimation of the step-size (using estimation of the objective function curvature), the extrapolation scheme leads to a first algorithm with strong theoretical convergence guarantees.

* A second main contribution is an optimization algorithm that assumes that once a candidate support of $w$ is known, a second-order-like method can be applied to minimize $f(w)$ restricted to the support. The approach is rather heuristic and  alternates between projected gradient and projected Newton-like update. The expectation is to achieve faster convergence using the second-order method. The algorithm comes with theoretical super-linear convergence guarantees.

* The two algorithms are supported by solid empirical evaluations. Reported results show that accelerate projected gradient (APG) is faster than PG. By incorporating the Newton-like update in APG leads to better computation performance. This demonstrates the effectiveness of the methods.

* The conducted empirical evaluations were restricted to projected gradient methods. However solving $\ell_0$ sparse problems has seen a trend of research works. The paper lacks to relate and compare proposed approaches to methods beyond projected gradient.

---

> ### Author Response · Authors · 2022-08-02
> **Response to vhqu (Part 1 of 2)**
>
> > In algorithm 1, how often $w^k$ and  $w^{k-1}$ may not activate the same support?
>
> This depends on a number of factors including the quality of the initialization, the value of $s$, and the degeneracy of the stationary point. Due to these, it is difficult to determine the frequency at which same subspace activation of consecutive iterates will occur in theory.
> However, to provide some samples, for the problems we ran experiment on, now we have also provided such information in Appendix C.2.
> We can see from the tables that at least in our experiments, the extrapolation takes place very frequently for our APG algorithm.
> The information of APG+ and APG-LL+ shows smaller percentages of extrapolation being executed, but this is due to the fact that after Unchanged $\geq S$, the algorithm stops attempting extrapolation.
>
>
> > In practice, how  $\alpha_{\min}$ and  $\alpha_{\max}$ are set in Algorithm 1?
>
> All the parameters setting of our experiments are shown in Appendix B. We set in particular $\alpha_{\min}=1$ and $\alpha_{\max}=100$ throughout the experiments.
>
> As mentioned in our paper (Lines 227-230), current extrapolation strategies require stepsizes to be confined in [0,1]. We set the maximum value of the traditional extrapolation, i.e. the unit stepsize, to be the lower bound of our stepsize. That is, we set $\alpha_{\min}=1$. For $\alpha_{\max}$, we can set it to be arbitrarily large, but to avoid drastic changes in the objective which can compromise the overall convergence of the algorithm one might not want to make it too huge. In general, $\alpha_{\min}$ could be rather small and $\alpha_{\max}$ could be rather big, and they just serve as a safeguard to prevent pathological cases.
> In our experiments, we can also set $\alpha_{\max}$ to larger values and will still obtain good numerical results (actually this upper bound is barely activated).
>
> > What are the typical values of $S$ in Algorithm 2? How  $S$ influences convergence? Activating the same sparsity support for a certain number  of iterations is not a certificate that the "correct" support is identified not a convergence in the support is attained.
>
> Similar to our response to the previous comment, the value of $S$ we used for our experiments ($S=5$) is also recorded in Appendix B.
>
> But indeed, activating the same support for a certain number of consecutive iterations does not always indicate that the right support is already identified. However, this is the most practical way to conjecture so that we can think of, as making certain that identification has already taken place requires knowledge of the exact location of the limit point (while if we already know where this point is, there is surely no any need for using any algorithm to solve the optimization problem).
> We have found, however, that the algorithm is not quite sensitive to the choice of $S$ in order to obtain stationary points/local minimum. If $S$ is set to a lower value, then it can be viewed as providing good “warm starts” for our algorithm that will drive the residual to a lower value with fewer iterations, but at a possibly higher computational cost (depending on the value of $s$). Certainly, we can also set some criteria to allow for changing values of $S$ in our algorithm, and this is a subject of our near-future work along our software development for our algorithm.
>
> > The conducted empirical evaluations were restricted to projected gradient methods. However solving  sparse problems has seen a trend of research works. The paper lacks to relate and compare proposed approaches to methods beyond projected gradient.
> > How the PG methods perform compared to other existing approaches to deal with  $\ell_0$ sparsity problem (such as MIP? greedy methods...)?
>
> Thanks for the suggestion. We have now added discussion about existing works on iterative hard thresholding and greedy methods in Lines 48-50 and Lines 57-62 following the references suggested by reviewer TQ21.
> In terms of numerical comparison, we have compared with a greedy method GraSP [2] that is a popular greedy method in Appendix C.6. We see that indeed this method is fast at first, but it only works for small-scale problems and is prohibitive for larger data due to its high memory cost. We also observe that for medium-scale problems, it is slower than our method.
> Comparison with MIP has already been done in [10], and their observation is that MIP alone is even slower than PG. Although MIP can be significantly accelerated by using PG as a warmstart mechanism, our method can also be combined with MIP in the same manner, and since our methods are faster than PG for generating a warmstart point, the combination with MIP of our methods will also certainly be faster than the combination of PG and MIP in [10].

---

> > ### Comment · Reviewer_vhqu · 2022-08-08
> > **Response to the authors**
> >
> > I thanks the authors for the time and effort in addressing my comments and providing clarifications. I appreciate the authors' effort in discussing related approaches and providing additional experimental results.

---

> ### Author Response · Authors · 2022-08-02
> **Response to vhqu (Part 2 of 2)**
>
> > As the prediction performances in Table are almost similar for all methods but the methods differ in computation time, do the same observations hold if one relaxes the stopping criterion $\epsilon$?
>
> We have now conducted some new experiments to see how the prediction performance changes with $\hat\epsilon$ in Appendix C.5.
> We can see that $\hat \epsilon = 10^{-5}$ almost always provides the best prediction performance no matter which algorithm is considered.
> This suggests that indeed a tight enough stopping criterion is desirable in such tasks.
> On the other hand, for reasonably relaxed values of $\hat \epsilon$,
> we still observed similar difference on computing times between non-accelerated and accelerated algorithms.
> From a theoretical perspective (Theorem 2.1), however, the prediction performances obtained from the earlier iterations may not correspond to local minimum/stationary points and our expeirments in Appendix C.5 provides some empirical supports for this claim.

---

### Author Response · Authors · 2022-08-02
**Summary of changes**

We sincerely thank all reviewers for their careful evaluation of our work and their invaluable suggestions and comments.
We have tried to address most comments of the reviewers in the revision, and all major updates are highlighted in red (there are wording changes that are mainly for fitting within the page limit, and are thus not highlighted).

We will individually reply to each reviewer's comments to address their specific comments, and here we first summarize the major changes in our revision.
In our response, when we refer to a line number, an equation number, a reference, a theorem, etc, we are referring to the number in the new revision unless otherwise specified, while when we quote the reviewers' comments, those are numbers in the original submission.

1. Additional experiments in Appendix C: several reviewers suggested various additional experiments, and due to that the space limitation is still 9 pages during the rebuttal and discussion period, we are putting all new experiments in the appendix. If our paper is accepted, with the extended 10-page limit, we will move suitable experiments to the main text in the camera-ready version.
2. We are able to prove a stronger result in the first claim of Theorem 2.2 due to reference [24]. In particular, $O(k^{-1})$ is improved to $o(k^{-1})$ in equation (10).
3. Revised statements of Theorem 3.2(a) and (b); an additional new result is provided in Theorem 3.2(c). Assumptions are relaxed such that we do not require $T_{PG}^{\lambda}$ to be a singleton at $w^{*}$.

---

### Meta-Review · Area_Chair_cX4y · 2022-08-30

**Recommendation:** Accept
**Confidence:** Certain

**Metareview:**

The paper studies sparsity-constrained optimization problems, in which the goal is to optimize a convex, Lipschitz function subject to and L0 constraint. It studies two algorithms: one based on extrapolation within the current support, and another which applies a second order method within the current support. The paper's theoretical results include (i) global convergence results for projected gradient methods with a locally linear rate, (ii) fast (linear, superlinear) rates for the proposed accelerated methods. Experiments show that the proposed methods indeed outperform projected gradient, and enjoy very fast convergence once the correct subspace is identified. Reviewers all note that the paper's theoretical results improve over the existing literature on convergence for sparsity-constrained problems. After interacting with the authors, the reviewers converged to a recommendation to accept the paper; the AC concurs.


**Award:**

No

---

### Decision · Program_Chairs · 2022-09-14

Accept